# Synthesis, Performance, Mechanism: A Hyperbranched Phase Reverse Nano-Demulsifier for Condensate Emulsion

**DOI:** 10.3390/molecules28237692

**Published:** 2023-11-21

**Authors:** Lei Liang, Chao Su, Yujia Xiong, Lei Wei, Congyue Gu, Haifeng Ye, Qinghua Xiao, Xingyu Luo

**Affiliations:** 1CCDC Geological Exploration and Development Research Institute, Chengdu 610051, China; shuc_dyy@cnpc.com.cn (C.S.); xiongyj_dyy@cnpc.com.cn (Y.X.); gucy_dyy@cnpc.com.cn (C.G.); yehf_dyy@cnpc.com.cn (H.Y.); xiaoqh_dyy@cnpc.com.cn (Q.X.); jerryleon1224@outlook.com (X.L.); 2Sichuan Hengyi Petroleum Technology Service Co., Ltd., Chengdu 610051, China

**Keywords:** oil and gas field development, condensate emulsion, oil–water separation, phase reverse demulsifier, hyperbranched nanomaterials

## Abstract

Organic amine and nanosilica were combined to create a nano-demulsifier, which was employed in the oil–water separation process of a condensate emulsion. The nano-demulsifier has the structure of hyperbranched polymers and the skeleton structure of hyperbranched nanomaterials, and displays the demulsification impact of organic amine polymers as well as the synergistic effect of nanomaterials. This nano-demulsifier has the potential to drastically reduce the quantity of condensate demulsifiers utilized in the gathering station. The dehydration rate of the condensate lotion in the gas gathering station can reach more than 95% only at a concentration of 1.0 wt.%. Its application can significantly increase the separation efficiency of the condensate emulsion as well as the quality of condensate oil. It has a positive impact on cost reduction and efficiency in gas well production. The mechanism of action of the demulsifier was also studied, and the results show that the demulsifier is a phase reverse demulsifier.

## 1. Introduction

The development of gas fields is often accompanied by the production of condensate oil [1,2,3]. It is highly challenging to separate the condensate oil from the emulsion [4,5]. During the production process, a significant amount of surfactant liquid is injected into the gas well [6,7]. As a result, it forms a complex emulsion system between well-entering fluids and condensate oil [8,9]. This not only has an impact on the quantity and quality of condensate production, but also poses a significant obstacle to the regular functioning of purification plants and gas field production.

Nowadays, regular demulsifiers are used in oilfields. For example, Zhang et al. disclosed a hyperbranched polyglycidyl methacrylate as an O/W type emulsion demulsifier [10], with a dehydration rate of 90% at a concentration of 2000 mg·L^−1^. Xu et al. synthesized a hyperbranched polymer inverse demulsifier using methyl acrylate and 1,3-propanediamine [11]. At a concentration of 40 mg·L^−1^, a dehydration rate of 91% was achieved. The aforementioned research has positive impacts on demulsifying crude oil in general, but its applicability to condensate oil still requires research. However, in terms of the demulsifier mechanism, the demulsification for O/W-type emulsions is typically treated using a reverse demulsifier. The demulsification is generally only successful when the molecular weight of the demulsifier is greater than that of the natural emulsifier. When an inverse demulsifier is used as an emulsifier in an oil–water mixture, an inverse emulsion is formed. In terms of the polymer structure, hyperbranched polymers have the following advantages over regular polymers: a highly branched three-dimensional structure, more active sites, good solubility, strong heat stability, and high chemical stability. Because of these advantages, hyperbranched polymers have a wide range of applications, including in membrane separation, drug administration, tissue engineering, and so on [12,13].

Differing from crude oil demulsification, a condensate emulsion is a rare special emulsion layer used in demulsification in oil fields [14]. Its genesis is complex and changeable, and the molecular weight of the condensate is generally small, which makes it easier to emulsify the emulsion [15,16]. The selection of the demulsifier becomes difficult. Some studies have disclosed inventions related to condensate demulsifiers. However, the focus was on their application in condensate-producing sour gas fields. Large amounts of sulfonates and dispersants were added to the formulations, and they were less resistant to acidity and salt. The increase in the content of silicone surfactants in the condensate demulsifier increases the silicon content of the condensate, thereby reducing the condensate quality [17,18]. In addition, the demulsifier formulations have variable and complex components, and generally have poor stability [19].

In order to solve the problem of condensate demulsification, a hyperbranched phase reverse nano-demulsifier for condensate emulsion was prepared. It was used in the oil–water separation process of condensate emulsions in gas fields. The prepared condensate demulsifier has a good demulsification effect and can significantly improve the oil–water separation efficiency of condensate emulsions.

## 2. Results and Discussion

### 2.1. Characterization

Infrared spectroscopy analysis was conducted for the raw material nanosilica and the prepared product, HND. The Fourier transform infrared spectra are shown in Figure 1.

From Figure 1, it can be seen that the nanosilica raw material mainly contains the two characteristic absorption peaks at 3400 cm^−1^ (O−H) and 1100 cm^−1^ (Si−O) [20]. In the prepared product, the two peaks of HND were weakened, and the characteristic absorption peaks of C=O (1700 cm^−1^) and C−N (1300 cm^−1^) appeared [21,22]. Combined with the C−H characteristic absorption peaks at 2800 cm^−1^, this indicates stable binding of various raw materials [23]. HND was successfully obtained according to the preparation method.

Further observation of the microstructure of HND was carried out using a transmission electron microscope. Figure 2 shows the transmission electron microscope image of an HND sample.

In Figure 2, it can be clearly observed that the efficient polymerization of nanoparticles and polymers resulted in a hyperbranched network polymer. Research has shown that hyperbranched structures are beneficial for improving the demulsification performance of demulsifiers [24,25,26,27]. The prepared nano-demulsifier has a minimum size of about 100 nm. The synthesized product is a composite nanomaterial. The material consists of spherical silicon nanoparticles, with the surface modified with N-(2-aminoethyl)-3-aminopropyltrimethoxysilane (follows from the HND synthesis protocol), immobilized in a prepolymer matrix. The composite material has a branched topology. In addition to the demulsification performance of its organic polymer part, the nanostructure of the material will also improve its demulsification performance to a certain extent.

### 2.2. Performance

The demulsification ability of HND in the simulated emulsion was evaluated. Figure 3 shows the demulsification performance of HND under different demulsification situations.

The demulsification temperature has a considerable impact on the demulsification effect. Generally, the higher the temperature is, the worse the stability of emulsion is, so it is more conducive to oil–water separation [28,29,30]. The temperature curve in Figure 3a also confirms this point. In the curve, the dehydration rate of HND with a 1.0 wt.% concentration in the simulated emulsion is more than 90% within the test temperature range.

Generally speaking, the longer the demulsification time is, the higher the oil–water separation efficiency of the emulsion will be [31,32,33]. In the actual production process, all stages of production are continuous, so it is necessary to choose an appropriate demulsification time to increase production efficiency. According to the demulsification time curve in Figure 3a, 60 min after HND was added to the emulsion, the dehydration rate could reach more than 99%.

After analyzing the demulsification temperature and time of HND, the demulsification performance of the simulated condensate emulsion was studied at 40 °C for 1 h. As can be seen from the results in Figure 3b, the dehydration rate of the simulated emulsion shows a slow decline trend after increasing within the tested concentration range. At a concentration of 0.1 wt.%, the dehydration rate is more than 94%, which shows that HND can show a good demulsification effect in the simulated condensate emulsion at a very low concentration.

### 2.3. Applications

Next, we take the on-site condensate emulsion as an example to analyze the application possibility of HND in actual production. Figure 4 shows the demulsification performance between HND and gathering station demulsifier (GSD) at different concentrations.

In the experiment, 100 mL of a condensate emulsion from a gas field’s gas gathering station was added into a centrifuge tube, and HND with different concentrations and the existing demulsifier (GSD) from the gas gathering station were added. Figure 4 shows the change in demulsification performance with concentration. The experimental results show that the dehydration rate of HND for the condensate emulsion is higher than that of GSD within the test concentration range. Especially at a lower concentration, according to relevant standards, the dehydration rate of the condensate at room temperature (25 °C) needs to be above 85%. In the actual production process, the concentration of condensate demulsifier added to the gas gathering station is about 1.0 wt.%, meeting the needs of emulsion changes. Under the production conditions during testing, HND only requires a concentration of 0.1 wt.% to achieve the demulsification performance of GSD at 1.5 wt.%. Therefore, HND can greatly reduce the total amount of on-site agents added.

Subsequently, we also analyzed the separation effect of HND and GSD on the condensate emulsion. Figure 5 shows the demulsification of the condensate emulsion after adding demulsifiers for 24 h. The concentration of HND added in the centrifuge tube from left to right is 0.1 wt.%, 0.5 wt.%, 1.0 wt.%, 1.5 wt.% and 2.0 wt.%, respectively.

By analyzing the experimental results in Figure 5a,b, the proportion of each layer of condensate after the demulsification of the condensate emulsion is summarized in Table 1. The main components were the condensate (the top layer), displacement (the middle layer), and water (the bottom layer).

It can be seen from Figure 5 that the on-site condensate emulsion has a high degree of emulsification, which is milky white as a whole. After demulsification, the condensate emulsion can be divided into three layers, namely the condensate (the top layer), displacement (the middle layer), and water (the bottom layer). After adding two demulsifiers, the emulsion showed different separation degrees. Comparing Figure 5a,b, it is not difficult to see that the oil and water layers of the emulsion after demulsification with HND are clear and transparent, and the interface of the intermediate layer is clear. The thicker the oil layer, the thinner the intermediate layer, and the clearer the water layer, the better the separation effect is reflected.

The experimental results show that the prepared nanodemulsifier, HND, has good demulsification performance for the condensate emulsion, and can effectively solve the problem of difficult oil–water separation of condensate emulsion in the process of oil and gas field exploitation.

### 2.4. Mechanism

Polarizing microscope (PM) pictures of condensate emulsions were taken to explore the action mechanism of HND. Figure 6 clearly shows the difference in emulsion particle size before and after demulsification. Before adding the demulsifier, the emulsion particle size is small and evenly dispersed, and the system is rather stable. The demulsifier replaces the original oil–water interface facial mask after its insertion, and the droplets continue to coagulate and get larger. The interface becomes uneven, taking on an ellipsoidal shape, and the droplets vanish. This shows that the demulsifiers’ demulsification method is phase reverse demulsification.

The action mechanism of the demulsifier was explored by measuring the Zeta potential of the condensate emulsion before and after the treatment of HND. Table 2 shows that the initial potential of the emulsion is −52.59 mV; at this moment, the electronegativity of the emulsion particles is strong, indicating mutual exclusion. As a result, the emulsion system is extremely stable. After adding HND, the Zeta potential of the emulsion grows continually while the absolute value declines. Demulsifiers can replace surfactants at the oil–water interface and neutralize the surface charge of oil droplets, thereby weakening electrostatic repulsion. The unstable emulsion droplets clump together to create giant oil droplets, which ascend to the liquid’s surface. The droplets are simpler to aggregate, detach from the emulsion, and rise to the upper layer of the liquid surface as the demulsifier dosage increases.

## 3. Experimental Method

### 3.1. Materials

Nanosilica was purchased from Chemkey (Shanghai, China). 1,3-propanediamine and methyl acrylate were purchased from Aladdin Inc. The simulated condensate emulsion was prepared with 100 g of gasoline and 1 L of water (3000 r/min, 30 min). The station condensate emulsion was provided by Petrochina Changqing Oilfield Company (Xi’an, China). Other reagents in the experiment were commercially available analytical reagents (AR).

### 3.2. Preparation

In the reactor, 50 mL of anhydrous ethanol and 0.1 g of sodium ethoxide were added, respectively. After dissolution, 0.1 mol (7.4 g) of 1,3-propanediamine and 0.2 mol (17.2 g) of methyl acrylate were added. The bright yellow prepolymer was obtained by reacting at 80 °C for 8 h. We dispersed 1.0 g of hydrophilic nanosilica and 0.1 g of N-(2-aminoethyl)-3-aminopropyltrimethoxysilane (KH-792) in 50 mL of anhydrous ethanol, adjusted the pH to 4 with 1.0 mol of HCl, and reacted at 60 °C for 2 h to obtain a dispersed modified body. We then dissolved the above prepolymer and 0.1 g of sodium ethoxide in 50 mL of anhydrous ethanol, and added it to the dispersed modifier. After heating to 80 °C and reacting for 12 h, the orange yellow production obtained was the hyperbranched nano-demulsifier (HND).

### 3.3. Measurements

The structure of the materials was characterized by means of a Fourier transform infrared spectrometer (FTIR, Nicolet iS5, Thermo Scientific, Waltham, MA, USA) and a transmission electron microscope (TEM, JEM-2100F, JEOL, Tokyo, Japan). The dehydration rate (water removal efficiency) of condensate oil was measured using a water in oil analyzer (T-BD5-MS1204, Bigdipper, Shenzhen, China).

## 4. Conclusions

A kind of hyperbranched phase reverse nano-demulsifier has been synthesized, and it has the dual advantages of hyperbranched polymers and nanomaterials. During the demulsification process of simulated condensate emulsion and on-site condensate emulsion, both showed good demulsification performance and a good oil–water separation effect. HND is therefore a highly efficient condensate demulsifier.

The stronger the market competitiveness of HND compared to existing products, the more it can meet the production demand of gas wells at extremely low concentrations. This greatly reduces the storage space and dosing frequency of reagents in the gas gathering station and improves production efficiency. It is also necessary to calculate the industrialization cost, measure the balance point between price and concentration, and calculate the economic benefits of its application.

HND has the characteristics of a general phase reverse demulsifier, and in addition to its use in oil and gas field production, its application in other demulsification fields can also be explored in later research.

HND is a kind of phase reverse demulsifier. The demulsification mechanism is to reduce the electrostatic effect between emulsion particles and destroy the stability of emulsion, so as to achieve the purpose of oil–water separation.

## Figures and Tables

**Figure 1 molecules-28-07692-f001:**
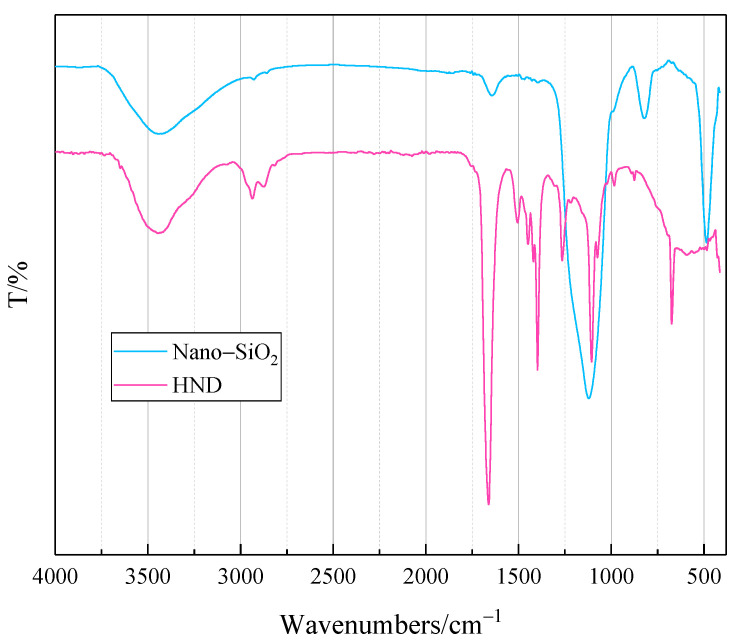
Infrared spectra of the materials.

**Figure 2 molecules-28-07692-f002:**
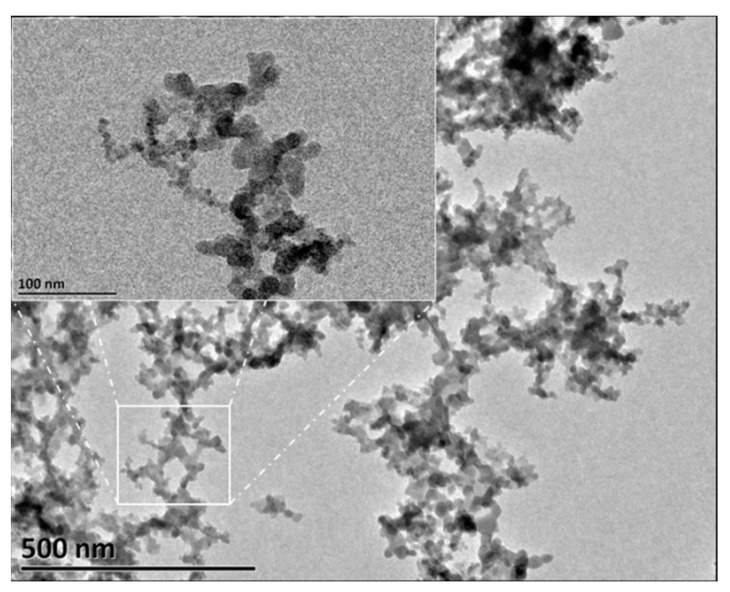
Transmission electron microscope image of HND.

**Figure 3 molecules-28-07692-f003:**
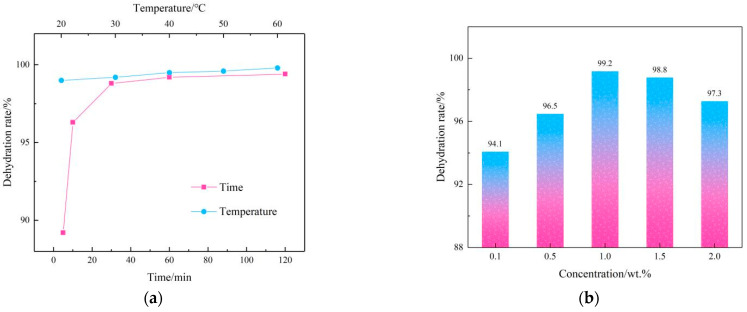
Demulsification performance of HND on a simulated emulsion under different conditions ((**a**) demulsification time and temperature, (**b**) demulsifier concentration).

**Figure 4 molecules-28-07692-f004:**
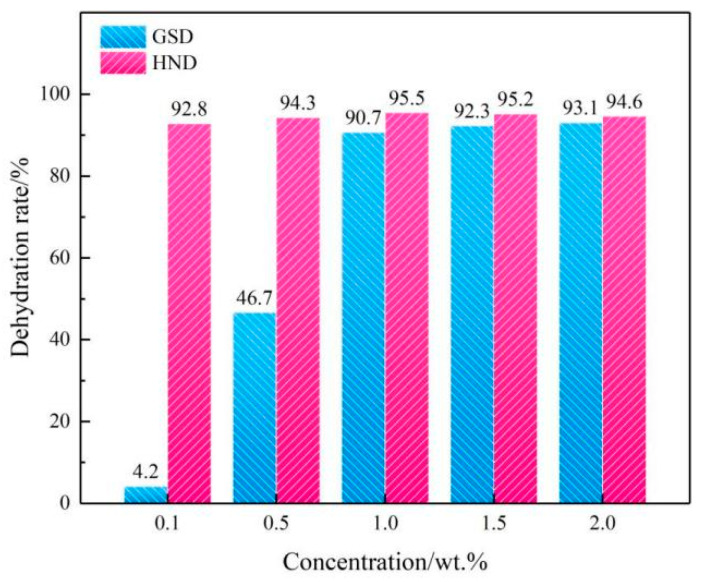
Demulsification performance of HND and GSD for the condensate emulsion.

**Figure 5 molecules-28-07692-f005:**
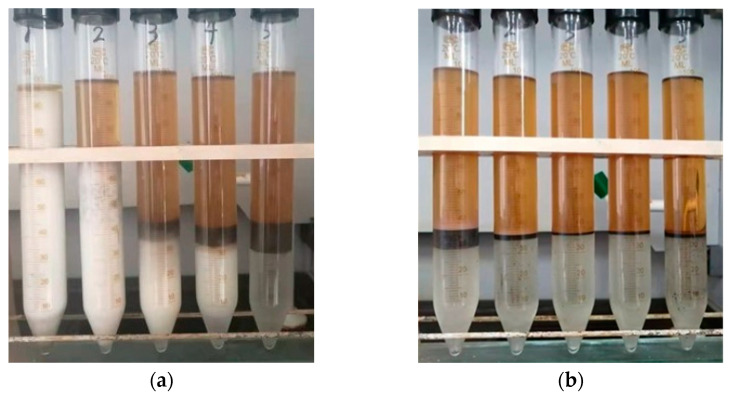
Demulsification of condensate emulsion after adding demulsifiers for 24 h ((**a**) GSD, (**b**) HND).

**Figure 6 molecules-28-07692-f006:**
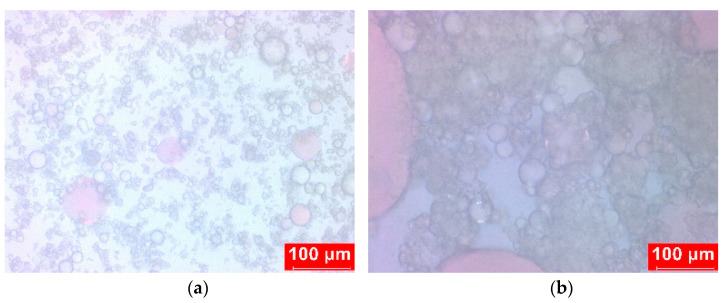
PM images of condensate emulsion ((**a**) before demulsification; (**b**) after demulsification).

**Table 1 molecules-28-07692-t001:** Proportion of each layer of condensate after demulsification.

Concentration /wt.%	Top/mL		Middle/mL		Bottle/mL	
	GSD	HND	GSD	HND	GSD	HND
0.1	2	62	96	11	2	27
0.5	28	63	69	3	3	34
1.0	57	63	8	1	35	36
1.5	58	63	7	1	35	36
2.0	58	64	12	2	30	34

**Table 2 molecules-28-07692-t002:** Zeta potential of the condensate emulsion treated with different concentrations of HND.

Concentration /wt.%	Zeta Potential /mV
0	−52.59
0.5	52.35
1.0	−49.91
2.0	−43.45

## Data Availability

Data are contained within the article.

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
