# Peer review of "Synthesis, Performance, Mechanism: A Hyperbranched Phase Reverse Nano-Demulsifier for Condensate Emulsion"

_molecules, 2023, doi:10.3390/molecules28237692_

Round 1

Reviewer 1 Report

Comments and Suggestions for Authors

The review of the manuscript entitled: “Synthesis, Performance, Mechanism: A Hyperbranched Phase Reverse Nano-Demulsifier for Condensate Emulsion”.  The authors completed very interesting work. The work is in the scope of the journal. By responding to the following comments, the work can be ready for publication:

1.      The kind of emulsion for separation should be mentioned. Oil in water or water in oil. The term “reverse” in the title cannot cover this comment as the definition in not fixed

2.      What is the reason for the efficiency of 95% by 1wt.%?

3.      The results are limited for condensate oil. However, the used material can be used for any kind of oil.

4.       “conventional crude oil demulsification” is not correctly presented. Now day, the regular demulsifiers are used in the oilfields.

5.      In the Introduction section, it is recommended to mention the stability of emulsion in the presence of asphaltene for petroleum emulsions considering the content of water in the system. For this purpose, the next reference can be used in the revision stage: https://doi.org/10.1080/10916466.2022.2049819

6.      What was the devise for efficiency measurement? In this area, we have two topics: water removal efficiency and salt removal efficiency. It should be correctly evaluated.

7.      The model development by response surface methodology is recommended for next works.

Reviewer 2 Report

Comments and Suggestions for Authors

The current manuscript by Liang et al represents a short experimental study on the demulsification ability of silica particles with a branched amine polymer. The study is short and concise, reaching a nearly complete separation of the petroleum emulsions used in the study. This is the reason I recommend its publication after a major review:

-The authors lack information on the demulsification mechanisms and properties of SiO2-amine complexes from literature. This seems to be best done in the presence of cationic surfactants. I highly recommend that the authors include additional literature to emphasize the difference between cationic surfactants and the hyperbranched polymer here. Here is an example review article considering the topic: https://doi.org/10.1007/s13202-019-0685-y.

-Please describe in the methods, the procedure for assessing the stability of the emulsions and their origin/preparation.

-Please correct Figure 7 to show zeta potential, not phase plots. What are the phase plots used for?

-The images from the optical microscopy suggest what is happening with the morphology, but can hardly be used to access the mechanism directly. Please provide more details on the emulsion content prior to use: ingredients, interfacial tension, viscosity of the oil phase, and concentration of oil, particles wetting by the oil. Then I recommend to add a discussion on the mechanism based on surfactants, compared to your polymer. 

Comments on the Quality of English Language

Minor editing of English language required

Reviewer 3 Report

Comments and Suggestions for Authors

The topic of this paper is solving the problem of desalting and dehydrating oil emulsions using demulsifiers to increase the recovery efficiency of gas-oil wells. Creation of effective properties of reagents - demulsifiers for the destruction of oil emulsions is an urgent and practically significant task. The authors propose to combine an organic polymer and nanosilica to create a new nanodemulsifier with increased efficiency for separating oil and water condensate emulsions.

Questions and Comments

1.           In part Abstract (line 10) - “skeleton structure of nanomaterials” – what the authors mean. It is necessary to clarify the structure of the material  

2.           In part Introduction:

-                the problem of separating oil and water in a condensate emulsion to improve the efficiency of gas and oil production is not presented consistently enough;

-                the advantages of hyperbranched demulsifiers compared to surfactants are not clearly positioned;

-                there is no consistent justification for the choice of additives and justification for the addition of silicon-containing components to improve the properties of the demulsifier;

-                there is no clear statement of the aim of the study.

3.           In part Experimental Method:

- 2.1 Materials: Ñ€hrase «Other reagents in the experiment were commercially available analytical reagents» requires an explanation indicating the list of reagents and their purity/grade;

- sodium ethanol is an unfortunate name. Correct sodium – ethoxide;

- nano-demulsifier (HND) is a hyper-branched nano-demulsifier? In this case, the abbreviation must be deciphered completely.

4.     In part Results and Discussion:

- It is necessary to provide a reaction scheme for the synthesis of HND, indicating the structural unit of the synthesized prepolymer and the proposed structure of HND;

- Figure 1 – make a correction to the labels of the coordinate axes in accordance with the presentation rules - T, % or T (%); Wavelength, nm or Wavelength, nm. The axis labels in Figures 3 and 4 also require correction;

- Figure 7 - axis labels are too small and unreadable;

- It is necessary to make a correct description of the IR spectra of compounds in accordance with the terminology of the method (correlation of bands or frequencies of stretching or bending vibrations of characteristic groups);

- The synthesized HND is not a hyperbranched polymer, unless this is proven by NMR spectroscopy data with calculation of the Degree of Pseudogeneration Branching. According to TEM data (Figure 2), the synthesis product is a composite nanomaterial. The material consists of spherical silicon nanoparticles d = 5-8 nm (clearly visible on TEM), surface modified with N-(2-aminoethyl)-3-aminopropyltrimethoxysilane (follows from the HND synthesis protocol), immobilized in a prepolymer matrix. The composite material has a branched topology. The authors need to once again reconsider their results and pay more attention to understanding the chemical processes of obtaining and the structure of the HND demulsifier;

- It is necessary to clarify the nature or classification of the substance Gathering Station Demulsifier used by the authors as a reference demulsifier;

- Has the influence of mechanical impurities (iron sulfide, clay particles, silt, etc.) that are present in oil at the interface and devoted to the strengthening of the film enveloping the water globules been assessed?

- The paragraphs 3.2 – 3.4 must be specified and presented in accordance with the rules of the journal

5.     In part Conclusions: Conclusions need to be supplemented with characteristics of the composition, particle size and morphology of the synthesized nanomaterial

6.     The References section contains 58% of references earlier than 2018. Thus, it is difficult to assess the current (over the last 5 years) state of research in the subject area of the article

Round 2

Reviewer 1 Report

Comments and Suggestions for Authors

The work was revised based on the comments. So it is recommended for publication.

Reviewer 2 Report

Comments and Suggestions for Authors

The recommendations following my previous revision were not entirely followed and the authors did not emphasize the difference in their hyperbranched polymer system compared to cationic surfactants with silica particles (where similar aggregates are obtained). 

Nevertheless, the article's improvement in demulsification in general is worth publishing, which is the reason I recommend its publication at this stage.

Reviewer 3 Report

Comments and Suggestions for Authors Dear authors, Your answers are satisfactory and the changes are sufficient.